# Collaborative Approach to Reach Everyone with Familial Hypercholesterolemia: CARE-FH Protocol

**DOI:** 10.3390/jpm12040606

**Published:** 2022-04-09

**Authors:** Laney K. Jones, Marc S. Williams, Ilene G. Ladd, Dylan Cawley, Shuping Ge, Jing Hao, Dina Hassen, Yirui Hu, H. Lester Kirchner, Maria Kobylinski, Michael G. Lesko, Matthew C. Nelson, Alanna K. Rahm, David D. Rolston, Katrina M. Romagnoli, Tyler J. Schubert, Timothy C. Shuey, Amy C. Sturm, Samuel S. Gidding

**Affiliations:** 1Heart and Vascular Institute, Geisinger, Danville, PA 17821, USA; mglesko@geisinger.edu (M.G.L.); asturm@geisinger.edu (A.C.S.); 2Genomic Medicine Institute, Geisinger, Danville, PA 17821, USA; mswilliams1@geisinger.edu (M.S.W.); igladd@geisinger.edu (I.G.L.); dcawley1@geisinger.edu (D.C.); akrahm@geisinger.edu (A.K.R.); tjschubert@geisinger.edu (T.J.S.); ssgidding@geisinger.edu (S.S.G.); 3Department of Pediatrics, Geisinger, Danville, PA 17821, USA; sge@geisinger.edu (S.G.); mcnelson@geisinger.edu (M.C.N.); 4Department of Population Health Sciences, Geisinger, Danville, PA 17821, USA; jhao@geisinger.edu (J.H.); dahassen@geisinger.edu (D.H.); yhu1@geisinger.edu (Y.H.); hlkirchner@geisinger.edu (H.L.K.); 5Department of Community Medicine, Geisinger, Danville, PA 17821, USA; mkobylinski@geisinger.edu; 6Department of Internal Medicine, Geisinger, Danville, PA 17821, USA; ddrolston@geisinger.edu (D.D.R.); tcshuey2@geisinger.edu (T.C.S.); 7Department of Translational Data Science and Informatics, Geisinger, Danville, PA 17821, USA; kmromagnoli@geisinger.edu; 8Geisinger Commonwealth School of Medicine, Scranton, PA 18510, USA

**Keywords:** familial hypercholesterolemia, prevention, primary care, implementation science, cholesterol screening

## Abstract

The Collaborative Approach to Reach Everyone with Familial Hypercholesterolemia (CARE-FH) study aims to improve diagnostic evaluation rates for FH at Geisinger, an integrated health delivery system. This clinical trial relies upon implementation science to transition the initial evaluation for FH into primary care, attempting to identify individuals prior to the onset of atherosclerotic cardiovascular disease events. The protocol for the CARE-FH study of this paper is available online. The first phase of the project focuses on trial design, including the development of implementation strategies to deploy evidence-based guidelines. The second phase will study the intervention, rolled out regionally to internal medicine, community medicine, and pediatric care clinicians using a stepped-wedge design, and analyzing data on diagnostic evaluation rates, and implementation, service, and health outcomes.

## 1. Introduction

Familial hypercholesterolemia (FH) is a hereditary cause of high cholesterol that leads to premature cardiovascular disease. Despite evidence-based guidelines for diagnosis and treatment, FH remains under-diagnosed and under-treated [1,2,3,4]. Individuals with a pathogenic genetic variant in an FH gene have triple the risk for atherosclerotic cardiovascular disease (ASCVD) compared with those without a variant at any low-density lipoprotein-cholesterol (LDL-C) level, presumably due to lifelong exposure to elevated LDL-C levels. The diagnosis of FH is often made in middle-aged adults, after experiencing premature ASCVD [5]. Event rates for an FH patient with prevalent ASCVD are 5-fold higher compared to those with no prior ASCVD [6]. Treatment beginning in adolescence lowers the risk for ASCVD before the age of 40 years from about 25% to <1% [3,7]. Diagnostic tools and effective medications to lower LDL-C exist. Prevention of ASCVD has been proven to be cost-effective. The care gap created by the lack of FH diagnosis is large.

Published focus groups conducted with individuals with FH and clinicians have found information technology (IT)-based algorithms to identify FH to be acceptable, appropriate, and feasible [8]. However, barriers exist to referring potential individuals with FH for clinical care including unfamiliarity of patients and non-specialty clinicians with the importance of an FH diagnosis, lack of experience with IT-based diagnostic strategies, concerns about genetic discrimination, psychological consequences related to a genetic diagnosis, gender/race/ethnicity, and time/cost burden [9]. IT-based strategies currently in use, and proposed, identify a large number of patients (1–2% of all patients in a given cohort) needing evaluation—more than can be accommodated into existing lipid specialty clinics [10,11,12]. Though FH care in an uncomplicated patient without ASCVD is relatively straightforward, (i.e., start a statin, monitor LDL-C levels, assess for side effects), lack of awareness among primary care clinicians regarding FH and discomfort with starting lipid lowering therapy in younger patients has contributed to this observed care gap. Primary care clinicians will need enhanced management skills and systems to successfully manage the projected caseload.

Implementation strategies can be used to improve the uptake of evidence-based guideline care for FH. Implementation science has been used to create a care delivery model that facilitates the relaying of genetic diagnostic information to at-risk patients [13]. Processes for returning genetic results for many disease-causing genes are in place, but this remains a work in progress given the diversity and complexity of the information related to the many diseases covered by the program. For FH, complicating factors are: explaining ASCVD risk, including the rationale for treating early in life, genetic diagnosis, medications, and the need for family screening when genetic testing is positive. Elements of an FH implementation care plan include: integrating primary care clinicians, specialty clinicians, and physician extender roles; providing patient and clinician education; initiating guideline-recommended care, screening of first-degree relatives; allowing for process improvement; growing expertise/satisfaction within the primary care workforce; and including evaluation tools for outcomes related to establishing an FH diagnosis.

This project is a collaborative approach to identifying and reaching every patient in Geisinger with familial hypercholesterolemia. It aims to improve diagnostic evaluation rates of FH by transitioning the initial screening for FH into primary care and initiating guideline-based management for individuals prior to the onset of ASCVD. This paper presents the protocol for the Collaborative Approach to Reach Everyone with Familial Hypercholesterolemia (CARE-FH) study. 

## 2. Materials and Methods

### 2.1. Overview

The primary outcome of the CARE-FH study is to determine if clinicians who receive an implementation strategy package (defined as a collection of targeted implementation strategies) are more likely to increase diagnostic evaluation rates for FH and if this is acceptable and sustainable within current primary care practice [14]. Implementation science frameworks have been shown to increase the generalizability of research findings [15,16,17]; therefore, the results of this study will be aligned with the Conceptual Model of Implementation Research (CMIR) framework. The evidence-based interventions are derived from two clinical practice guidelines: 2018 American Heart Association and American College of Cardiology (AHA/ACC) Cholesterol Guidelines and the 2018 *JACC* Genetic Testing Statement [1,18]. The implementation strategies will be designed to deploy these evidence-based interventions into clinical practice. Outcomes in domains of implementation, service, and health will be sought. Generalizability to other health care systems will be an additional key goal [19]. CARE-FH is registered with ClinicalTrials.gov (accessed on 8 March 2022), number NCT05284513.

The initial phase of the grant (R61 National Institutes of Health (NIH) mechanism) will focus on trial design, including the development of implementation strategies to deploy the evidence-based intervention. The implementation phase (R33 NIH mechanism) will study the intervention, using a stepped-wedge design, and analyze data on FH diagnostic evaluation rates, and implementation, service, and health outcomes (Figure 1). Practice randomization strategies will be used as part of the stepped wedge design and rollout.

### 2.2. FH Diagnostic Evaluation Program

This program for FH care is based on the current paradigm for notification regarding actionable genetic variants in MyCode Community Health Initiative (MyCode) a research program that links exome sequence data with electronic health record (EHR) data and returns pathogenic variants in clinically actionable genes to patient-participants (Figure 2) [13,20,21,22,23,24,25]. Evaluation is triggered by the notification of the patient and primary care clinician of an actionable variant. In CARE-FH, there will be four possible triggers: the presence of an FH variant from MyCode, cascade screening of family members identified in the MyCode program, identification of patients requiring an FH workup through applications of FH screening algorithms of EHR data, or presence of an LDL-C ≥ 160 mg/dL in a child. Notification will lead to scheduling a medical appointment to complete the FH diagnostic evaluation. Successful completion of the FH diagnostic evaluation, the primary outcome of this study, will include evidence the clinician has completed evidence-based FH diagnostic evaluation and management goals. The goals are defined as completing one of the following:-Used FH clinic note to document care-Added FH diagnosis on the problem list or used the Dutch Lipid Clinic Network score (DLCN) tool to exclude FH diagnosis-Used the FH smart-set (i.e., ordered a genetic test for FH)-Made a referral to the lipid clinic [12]-Initiate evidence-based lipid lowering medications

A patient will be considered to have FH if they have a positive genetic test for FH or have a DLCN > 8 and probable FH with a score of 5–8. Geisinger primary care sites are divided by geographic region (Central, Northeast, Western, or Geisinger Medical Center) and into several practice types. Adult-oriented primary care clinics are staffed by internists, family practitioners, and advanced practice professionals (physician assistants and nurse practitioners). Pediatric practices provide primary care and acute care at 22 clinics.

### 2.3. Potenital Implementation Strategies

Based on previous work [26], preliminary implementation strategies were developed for Aim 1 of this study (Table 1). These implementation strategies were mapped to the Expert Recommendations for Implementing Change (ERIC) compilation, a list of strategies generated by experts in implementation strategies, to improve their generalizability to future research studies [19]. Also, the implementation strategies were defined using the Proctor’s framework for specifying implementation strategies and these will be modified in Aim 1 of the study [27].

## 3. Specific Aim 1 (R61): Design a Clinical Trial to Assess Multi-Level Implementation Strategies for Improving FH Diagnosis in an Integrated Health System

### 3.1. Objectives and Work Plan by Team

#### 3.1.1. Implementation Science Team (ImpT)

The ImpT will meet bi-monthly to accomplish the objectives outlined below.

Identification of healthcare system level barriers.Tailor selected implementation strategies to meet the needs of the clinical implementation sites.Alpha testing of implementation strategy package into two preselected clinical implementation sites.Define a measurement of implementation outcomes.

Surveys and in-person contextual inquiries (defined as observations with interviews) with clinicians at community adult and pediatric practices will be conducted to understand barriers to the implementation of the proposed strategies. The implementation strategy package will be designed and reviewed in two deliberative engagement meetings with the Medical Science Team (MedT). The Informatics and Data Science Team (InfT) will design, build, test, and implement information technology (IT)-based strategies in partnership with the ImpT. The main outcome is the selection and design of the individual components of the implementation strategy package and definitions of implementation outcomes.

##### Survey

Prior to alpha-testing of selected implementation strategy package, Geisinger primary care clinicians will be asked to complete the 12-item validated survey, Acceptability of Implementation Measure, Implementation Appropriateness Measure, and the Feasibility of Intervention Measure [28], to assess readiness for adoption of the implementation strategy package. Descriptive analyses including means and standard deviations or median and inter-quartile ranges, depending on distribution, will be reported for each scale. Plots, including histograms and box plots, will be used to identify outliers. To examine trends across multi-level scales, each construct will be examined independently, as well as a composite adoption score. Scatter plots will be used to examine for any association between pairs of scales. Correlations between the various scales will be estimated using Pearson and Spearman rank-order correlation coefficients.

##### Contextual Inquiries

Human-centered design visual artifacts will be developed to synthesize and illustrate the current state of the process of FH identification at clinic sites. These artifacts may include service blueprints, journey maps, and/or experience maps as appropriate. These artifacts will serve as a baseline of the current workflow, clinician, and patient experience, and as a shared understanding to identify areas of opportunity to innovate and improve. As roll out will be by geographical region, (Central, Northeast, Western, and Geisinger Medical Center), covariates will be defined and summarized by region, individual clinic, and practice type (pediatric, internal medicine, family practice). Examples include: Cluster-level covariates such as the percent of the population meeting the definition of rural or underserved; Clinic-level covariates including size, clinician experience (including trainees); Clinician-level covariates such as demographics (age, sex, race), and clinician type.

##### Deliberative Engagement Meetings

Based on the findings of the survey and contextual inquiries, we will tailor the selected implementation strategies, using evidence-based techniques from human-centered design and implementation science. We will hold two deliberative engagement meetings with the MedT to develop and refine the multi-level implementation strategy package. One meeting will be held prior to the alpha test site roll-out; the second will further tailor the implementation strategy package to be used for the clinical trial (Aim 2).

#### 3.1.2. Medical Science Team (MedT)

The MedT will meet monthly to accomplish the objectives outlined below.

Partner with the InfT to revise content for EHR tools in the implementation strategy package and subsequent adaptations.Finalize FH care plan for adults and children.Finalize strategy for incorporation of genetics counselors and specialty referrals into a care plan.Finalize study timeline, including the schedule and sites for rolling out the implementation strategy package.Alpha test the implementation strategy package at one adult and one pediatric practice site.

Interviews performed by the ImpT with clinicians at adult and pediatric practices will be used to refine the structure for the rollout of the implementation strategies. Strategies to increase pediatric screening rates will be developed. The alpha test will occur at one adult and one pediatric clinic at the main hospital, staffed by a member of the MedT. All tools developed for the implementation strategy package will be tested, using principles from human-centered design. The alpha test will be further informed by interactions with end users to assess workflow as part of the human-centered design activities. The main outcome of the R61 phase of the study will be successful first rollouts.

#### 3.1.3. Informatics and Data Science Team (InfT)

The InfT will meet monthly to accomplish the objectives outlined below.

Partner with the MedT to develop content for building EHR tools for the implementation strategyProvide EHR support for the alpha test.Finalize the data analysis plan.Collect baseline outcomes data, including estimates of patient flow, for the clinical trial design.

The InfT will obtain the final study protocol and materials from the ImpT and MedT. These will then be embedded into Geisinger IT systems, with testing of the ability to present information (including patient flow) and extract data. Initial testing of EHR tools will occur in the EHR development environment. This utilizes an industry best practice for testing innovation and enhancements prior to full deployment, facilitates user interaction, allows feedback, and ensures the proposed interventions are integrated with clinician workflow. The primary outcome of Aim 1 for the ImpT will be the successful collection of study outcomes from the EHR to serve as baseline data for the usual care arm of the trial.

## 4. Specific Aim 2 (R33): Compare FH Diagnostic Evaluation Rates among Primary Care Clinicians Who Receive the Implementation Strategy Package versus Those Who Do Not

We will conduct a type 3-hybrid effectiveness-implementation trial using a stepped-wedge design to test the effectiveness of the implementation strategy package designed in the R61 phase of the project [29]. The rollout will occur by Geisinger geographic region: Central, West, Northeast, and Geisinger Medical Center. Clinics within each region will be randomized as to the order of receiving the implementation intervention. Data will be collected on the primary and secondary outcomes before and after the intervention to measure change. After each step, we will use an iterative review process to re-assess the fit of the multi-level implementation strategy package and make adaptations, such as repeat sessions for clinics with new trainees, as needed. This aim will examine implementation outcomes including adoption and penetration of an FH diagnosis program (Table 2). We hypothesize: That clinicians that receive the implementation strategy package will have an improved rate of diagnostic evaluation for FH compared to those in the usual care group.

### Data Collection and Analysis for Adoption and Penetration

All primary care clinics in the Geisinger system will be included in the study. Clinicians who have a patient who requires diagnostic evaluation for FH will receive an EHR-based notification. The clinician or designee will call the patient to discuss the next steps and schedule an appointment for an FH diagnostic evaluation (Figure 2). At the same time, adults, or parents of pediatric patients, will be notified by mail and telephone of the need for evaluation.

Data will be analyzed at the level of clinics, clinicians, and patients and stratified by the study phase. Bi-variate analyses will be used to assess differences in outcomes between the intervention and control periods. We will also follow the CONSORT 2010 [30] extension to stepped-wedge trials when reporting results. The percentages of FH diagnostic evaluations completed will be reported every six months. Follow-up will continue through the end of the study (48-months). Characteristics of clinicians and clinics will be summarized by exposure status to allow consideration of selection bias and lack of balance. FH diagnostic evaluation rate will be analyzed using a generalized linear mixed effects model [31,32], with completion of the FH diagnostic evaluation as a binary outcome, intervention, time, and their interaction as fixed-effects, cluster as random-effects, along with other covariates as predictors. The covariates will be selected by the advisory board based on previous evidence of the effect on FH diagnostic evaluation rates. A mixed effects model will be used to account for the hierarchical clustering structure of data (patients per clinician, clinicians per clinic). The model assumes that the intervention effect and time effects are common to all clusters, and that observations are equally correlated within clusters across time. In the primary analysis, adult and pediatric clinics will be analyzed separately however commonalities will be sought and, when appropriate, pediatric, and adult behavior will be compared. Of particular interest are differences in performance among adult and pediatric clinicians. Sample sizes were based on a power calculation considering a meaningful implementation strategy package as a 20% improvement in FH diagnostic evaluation rate and required a minimum of 10 clinics per region. Statistical analyses will be conducted using RStudio (Version 1.3.1093) [33]. *p*-values of less than 0.05 are considered statistically significant.

## 5. Specific Aim 3 (R33): Measure Implementation Success of an Organized FH Diagnostic Evaluation Program

Based on the CMIR [34,35] (Figure 1), this aim will examine implementation outcomes including acceptability, cost, feasibility, fidelity, and sustainability of an FH diagnostic evaluation program (Table 2). The following research questions will be explored:What is the acceptability of an FH diagnostic evaluation program across different demographic regions of the health system?How does the implementation strategy package fit (from Aim 2) within and between different clinic settings and patient populations and what adaptations were made?What are the costs to the healthcare system to implement and maintain an FH diagnostic evaluation program?

### 5.1. Data Collection and Analysis for Each Implementation Outcome

#### 5.1.1. Acceptability

Semi-structured interviews will be conducted with clinicians and patient participants for each of the four steps. Themes assessed in the interviews include empowerment of clinicians to screen and initiate management for FH, acceptability of the FH diagnostic evaluation program, knowledge, attitudes, and beliefs about FH, comfort with the FH diagnostic evaluation, acceptability of strategies for notification of potential FH patients and FH education, and adaptations that occurred. Sessions will be audio-recorded and transcribed verbatim. A thematic analysis approach will be utilized [36].

#### 5.1.2. Cost

Decision analysis and a simulation modeling approach will be applied and the costs of each step of implementation will be estimated utilizing a micro-costing approach, wherein, each component or resource use is estimated, and a unit cost is derived [37]. All models will be developed transparently using recognized methodological and reporting standards to support generalizability and rigor [38]. Decision analysis model development, simulation, and sensitivity analysis will be conducted using decision analysis software (e.g., TreeAge Pro, Palisade Decision Tools Suite) and/or Excel.

#### 5.1.3. Feasibility

Clinician adoption and penetration of completing the FH diagnostic evaluation and measuring the utility of the implementation strategy package will be measured through EHR data and semi-structured interviews. The analysis for the semi-structured interviews will follow the same structure that will be utilized for acceptability.

#### 5.1.4. Fidelity

Direct observation utilizing a checklist will occur at each clinic site after deployment of the implementation strategy package and observations of any adaptations that have been made to the process will be conducted. Guided by the framework for reporting adaptations and modifications (FRAME), we will report the following data: (1) when and how a modification was made, (2) whether the modification was planned/proactive or unplanned/reactive, (3) who decided to make the modification, (4) what is modified, (5) at what level of delivery the modification is made, (6) type or nature of context or content-level modifications, (7) the extent to which the modification is fidelity-consistent, and (8) the reasons for the modification [39].

#### 5.1.5. Sustainability

Clinician and patient participants will be surveyed to measure self-efficacy related to FH care and the institutionalization potential of the implementation strategy package. Descriptive analyses including means and standard deviations or median and inter-quartile ranges, depending on distribution, will be performed. Information collected will be reported and reviewed with the advisory board to discuss potential sustainability.

## 6. Specific Aim 4 (R33): Measure Patient-Related Outcomes after Implementation of an FH Diagnostic Evaluation Program

This aim will use the patient data collected and follow-up EHR data from subsequent FH care visits (Table 2). We hypothesize that: Service and exploratory health outcomes will be achieved more frequently by clinicians and their patients after the implementation strategy package is deployed to their clinic.

### 6.1. Data Collection and Analysis for Service and Health Outcomes

#### 6.1.1. Timeliness

For measurements looking at the time to completion of tasks, key dates will be the date of first FH notification, date of the clinic visit, and dates of supporting patient contacts such as genetic counseling and lipid specialty appointments. Visits with genetic counselors will be used to assist with the tracking of genetic testing and cascade screening of relatives. An initial visit to a lipid specialist will be used to determine the time for statin initiation, if not done at the primary care visit. The 2018 AHA/ACC cholesterol guideline will be used to define evidence-based indications and goals related to statin initiation [1]. Timeliness outcomes will be analyzed using time-to-event models, including the Cox Proportional Hazard regression model.

#### 6.1.2. Function

Data to be extracted from the FH Clinic Note and follow up assessments include the presence of a definite or probable FH diagnosis or used DLCN tool to exclude FH diagnosis, medication prescribing, medication-related side effects, lipid levels (to assess compliance with evidence-based care), genetic test ordering, return of genetic results, and initiation of cascade screening. We will use descriptive statistics and compare outcomes between intervention and control conditions every six months and with reference to the four steps.

## 7. Discussion

The overarching goal of the CARE-FH project is to increase critical elements of FH awareness within an integrated healthcare system and facilitate guideline-based care processes to identify individuals with FH. These would include for adults the ability for primary care clinicians to initiate the FH workup independent of a specialty clinician and triage patients in need of cardiovascular risk reduction management. Pediatric clinicians will perform FH screening by ordering lipid profiles for 9–11 years old or performing an evaluation of those identified through cascade screening. Implementation, Service, and Health outcomes measured in CARE-FH have a variety of meanings in different disciplines, therefore, we define their meanings for CARE-FH so that others have the ability to compare in the future.

The CMIR was selected as the guiding framework because it captures critical elements of the FH diagnostic evaluation program including defining the evidence for the creation of intervention, developing implementation strategies to deploy that evidence-based intervention, and put equal value on the measurement of implementation, health system, and individual (both patient and clinician) outcomes [34]. We chose the ERIC compilation to guide the development of our implementation strategies because these most closely matched the strategies identified through preliminary investigation [19]. The outcomes generated in the first phase of this study (R61) will utilize existing guidance in implementation science to map our implementation strategy to an existing compilation and define our strategies using recommendations for reporting strategies [27]. This will improve generalizability to other healthcare systems. In the second phase (R33 phase), we will add knowledge on how to deploy an implementation strategy package and measure the effect on implementation, service, and health outcomes.

Informatics tools also have the potential to improve outcomes of care. However, many decision support tools are designed without attention to clinician workflow leading to interruptions and increased clinician effort, ultimately limiting their impact. The CARE-FH study will utilize principles of user-centered design to better understand the clinician work process associated with FH diagnosis and care. The engagement with clinicians during the R61 phase will inform the type of decision support tools that are desired, but also understand where in the workflow they could be most effectively deployed to support clinician decision making and study goals. It is anticipated that different clinician types (e.g., primary care clinicians, specialists, pediatricians) will have different requirements to meet evidence-based guidelines for FH care. An understanding of these differences will be used to design informatics tools specific to the clinician type. The project will also use pilot testing and continuous improvement principles to optimize the design of any informatics tools. The clinical trial (R33 phase) will capture usability and clinician feedback to further refine the informatics tools. All informatics tools will be built using validated and, where available, certified informatics standards to enhance generalizability to other systems.

If successful, we expect this study to not only influence FH diagnostic evaluation rates at Geisinger but be generalizable to other health systems. Hopefully, higher diagnostic evaluation rates will lead to downstream acceptance of preventive lipid lowering care and lower risk for heart attacks for individuals and populations. In addition, the processes used herein may be generalizable to care efforts regarding other genetic conditions.

## Figures and Tables

**Figure 1 jpm-12-00606-f001:**
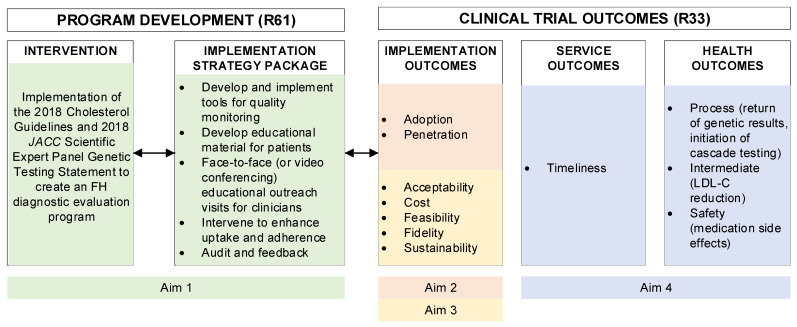
Conceptual model of implementation research framework tailored to the CARE-FH protocol. JACC, Journal of the American College of Cardiology.

**Figure 2 jpm-12-00606-f002:**
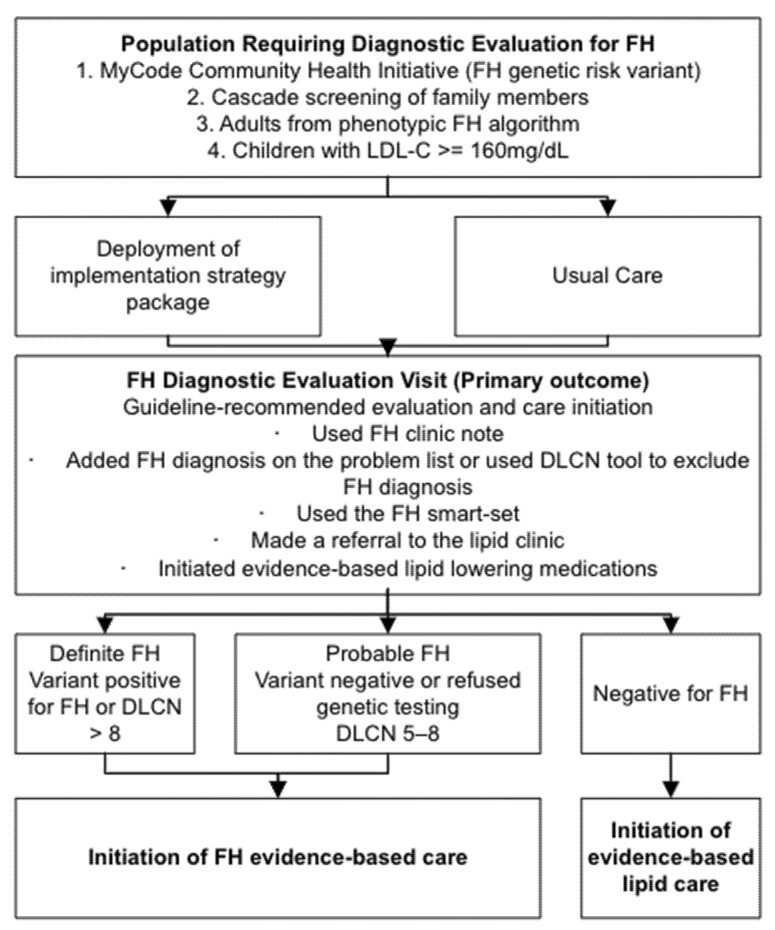
FH Diagnostic Evaluation Program. ASCVD, atherosclerosis cardiovascular disease; DLCN, Dutch Lipid Clinic Network score; FH, Familial hypercholesterolemia; Clinics will move from usual care to deployment of implementation strategy package during the stepped-wedge design.

**Table 1 jpm-12-00606-t001:** Potential implementation strategy package.

Name of Strategy *	Study Specific Definition	Actor	Action	Action Target
Develop and implement tools for quality monitoring	EHR tools to order labs, record results, and document FH care	ImpT, MedT, and InfT	Use EHR to record, order, and prescribe FH Care	Service and health outcomes
Develop educational materials	Education regarding guidelines for identification and treatment of FH	MedT and InfT	Create a CME course for clinicians about FH. Explore clinician workflow and educational needs to design novel focused educational interventions integrated within clinical workflows to support evidence-based care	MedT ready to train clinicians on FH
Conduct educational outreach visits	CME educational material for FH that is presented to each clinic	MedT and clinicians	Attend CME course on FH	Improve knowledge about FH
Intervene with patients to enhance uptake and adherence	Reach out directly to patients to recommend screening for FH	Clinicians and ImpT	Letter sent to the patient. Clinician schedules patient for appointment.	Patients diagnosed with FH from those at-risk
Identify and prepare champions	Clinical lipid champions	MedT	Identify and train lipid champions	Improved performance of study metrics, reduced costs
Stage FH care delivery model scale up	Develop the timeline for the stepped-wedge rollout to primary care	Leadership team	Notify practices of roll out and schedule education	Begin the trial
Audit and provide feedback	Provide aggregate level feedback to clinics on diagnosing FH	MedT, InfT, and clinical leadership	Report back to clinicians’ aggregate level data	Improve effectiveness of the FH Diagnosis Program
Advisory board review	Clinical trial protocol	Advisory Board	Provide feedback on the clinical trial regarding protocol, generalizability and ethical issues	Protocol revision based on feedback

* Mapped to the Expert Recommendations for Implementing Change (ERIC) compilation. Specification requirements for the implementation strategy will be tailored during aim 1 of CARE-FH. EHR, electronic health record; CME, continuing medical education; FH, familial hypercholesterolemia; ImpT, implementation science team; InfT, informatics and data science team; MedT, medical science team.

**Table 2 jpm-12-00606-t002:** Description of domains, aim, outcomes, construct measured, and data sources for phase two (R33).

Domain	Aim	Outcome	Construct Measured	Data Source
Implementation	2	**Adoption**	**FH diagnostic evaluation defined as completed of one of the following:** - Used FH clinic note to document care - Added FH diagnosis on the problem list or used DLCN tool to exclude FH diagnosis - Used the FH smart-set (i.e., ordered a genetic test for FH) - Made a referral to the lipid clinic - Initiate evidence-based lipid lowering medications	EHR, administrative data
		Penetration	Proportion of the primary care clinicians that completed the five components of the FH diagnostic evaluation compared to those that did not use it.	
	3	Acceptability	Clinician and patient satisfaction and self-efficacy with the implementation strategy package	Semi-structured interviews
		Cost	Cost to implement the implementation strategy package	Micro-costing
		Feasibility	Clinician adoption and penetration for completion of the FH diagnostic evaluation and measured utility of implementation strategy package	Semi-structured interviews and EHR data
		Fidelity	Documentation of adaptations to the FH diagnostic evaluation program	Checklist, direct observation
		Sustainability	Potential for institutionalization	Surveys, Advisory board consultation
Service	4	Timeliness	Time to: FH screen, completion of diagnostic evaluation, medication initiation	EHR, administrative data
Health		Safety	Medication-related side effects	
		Intermediate	LDL-C reduction	
		Process	Return of genetic result	
			Initiation of cascade screening	

EHR, electronic health record; FH, familial hypercholesterolemia; LDL-C, low-density lipoprotein cholesterol. Bolded is the primary outcome of the study.

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
