# Peer review of "Collaborative Approach to Reach Everyone with Familial Hypercholesterolemia: CARE-FH Protocol"

_jpm, 2022, doi:10.3390/jpm12040606_

Round 1
Reviewer 1 Report
Collaborative approach to reach everyone with familial hipercholesterolemia: CARE-FH protocol
The manuscript describes a protocol for the early detection of familial hypercholesterolemia. The authors detail a whole series of processes to implement, including analysis of the success of the implementation, possible differences between pediatricians and adult physicians, comparison between primary care physicians who implement the protocol and those who do not, and others. However, the manuscript remains in the description of material and methods, without results, which is of interest to the scientific community, especially when you want to apply a procedure of a certain complexity. Therefore, in my opinion, it is not publishable until they have results.
Minor comments
- DLCN consider definite FH with DLCN score >8, not >= 8, and probable with DLCN score 6-8, not 5-7 (2019 ES/EAS Guidelines for the management of dyslipidaemias: lipid modification to reduce cardiovascular risk, Eur Heart J 2020;41:111-188)
Page 3, line 117: define EHR (Electronic Health Record) in text. EHR Is defined only in table 1.
Author Response
The manuscript describes a protocol for the early detection of familial hypercholesterolemia. The authors detail a whole series of processes to implement, including analysis of the success of the implementation, possible differences between pediatricians and adult physicians, comparison between primary care physicians who implement the protocol and those who do not, and others. However, the manuscript remains in the description of material and methods, without results, which is of interest to the scientific community, especially when you want to apply a procedure of a certain complexity. Therefore, in my opinion, it is not publishable until they have results.
Response: We appreciate the reviewer’s opinion; however, this submission describes a study protocol which is within the journal’s scope. The manuscript follows accepted guidelines for protocol descriptions, and we received prior approval for the submission from the Special Issue Editor. We feel that it is important to share our implementation process so that our study can be replicated in the future.
Minor comments
- DLCN consider definite FH with DLCN score >8, not >= 8, and probable with DLCN score 6-8, not 5-7 (2019 ES/EAS Guidelines for the management of dyslipidaemias: lipid modification to reduce cardiovascular risk, Eur Heart J 2020;41:111-188)
Response: We appreciate the reviewer point out this error. We have updated the text and figure to reflect definite FH as >8 and probable 6-8.
Page 3, line 117: define EHR (Electronic Health Record) in text. EHR Is defined only in table 1.
Response: We thank the reviewer for pointing out that we forgot to define EHR in the text. We have updated the text accordingly.
Reviewer 2 Report
Overall/General Comments
In the present article, the authors describe a protocol for managing familial hypercholesterolemia (FH). Overall, an interesting and comprehensive approach regarding the early diagnosis and both timely and appropriate treatment of FH at the primary care level.
Specific Comments
- In page 4 the authors state: “A patient will be considered to have FH if they have a positive genetic test for FH or have a DLCN ≥ 8 and probable FH with a score of 5-7.” However, according to the Dutch Lipid Clinic Network diagnostic criteria probable FH is defined as having a score between 6-8. How was the 5-7 score selected as indicative of probable FH?
Author Response
In the present article, the authors describe a protocol for managing familial hypercholesterolemia (FH). Overall, an interesting and comprehensive approach regarding the early diagnosis and both timely and appropriate treatment of FH at the primary care level.
Specific Comments
- In page 4 the authors state: “A patient will be considered to have FH if they have a positive genetic test for FH or have a DLCN ≥ 8 and probable FH with a score of 5-7.” However, according to the Dutch Lipid Clinic Network diagnostic criteria probable FH is defined as having a score between 6-8. How was the 5-7 score selected as indicative of probable FH?
Response: We appreciate the reviewer point out this error. We have updated the text and figure to reflect definite FH as >8 and probable 6-8.
Reviewer 3 Report
This is a interesting detailed description of the clinical follow up of FH within the Geisinger health group. As such it has interest to the reader
Author Response
This is a interesting detailed description of the clinical follow up of FH within the Geisinger health group. As such it has interest to the reader
Response: We thank the reviewer for the support of our work.
Round 2
Reviewer 1 Report
If the editor considers that the manuscript falls within the objectives of the journal, I have no further comments.